# Insulin Pump Therapy Efficacy and Key Factors Influencing Adherence in Pediatric Population—A Narrative Review

**DOI:** 10.3390/medicina58111671

**Published:** 2022-11-18

**Authors:** Laura Mihaela Trandafir, Stefana Maria Moisa, Mihaela Victoria Vlaiculescu, Lacramioara Ionela Butnariu, Laura Otilia Boca, Maria Magdalena Leon Constantin, Paula Madalina Lupu, Crischentian Brinza, Oana Raluca Temneanu, Alexandru Burlacu

**Affiliations:** 1Pediatrics Department, University of Medicine and Pharmacy “Grigore T. Popa”, 700115 Iasi, Romania; 2“Sfanta Maria” Clinical Emergency Hospital, 700309 Iasi, Romania; 3Diabnutrimed Clinic, 020000 Bucharest, Romania; 4Faculty of Medicine, University of Medicine and Pharmacy “Grigore T Popa”, 700115 Iasi, Romania; 5Clinical Rehabilitation Hospital, 700661 Iasi, Romania; 6Institute of Cardiovascular Diseases “Prof. Dr. George I.M. Georgescu”, 700503 Iasi, Romania; 7Department of Mother and Child Medicine, University of Medicine and Pharmacy “Grigore T. Popa”, 700115 Iasi, Romania

**Keywords:** type 1 diabetes mellitus, adherence, insulin pump therapy, pediatric population, continuous subcutaneous insulin infusion

## Abstract

*Objective*: we aimed to highlight the state of the art in terms of pediatric population adherence to insulin pumps. This study intends to underline the significance of identifying and minimizing, to the greatest extent feasible, the factors that adversely affect the juvenile population’s adherence to insulin pump therapy. *Materials and methods*: articles from PubMed, Embase, and Science Direct databases were evaluated using the following search terms: adherence, pump insulin therapy, children, pediatric population, and type 1 diabetes, in combination with several synonyms such as compliance, treatment adherence, pump adherence, patient dropouts, and treatment refusal. *Results*: A better glycemic control is connected to a better adherence to diabetes management. We identify, enumerate, and discuss a number of variables which make it difficult to follow an insulin pump therapy regimen. Several key factors might improve adherence to insulin pump therapy: efficient communication between care provider and patients (including home-based video-visits), continuous diabetes education, family support and parental involvement, as well as informational, practical assistance, and emotional support from the society. *Conclusions*: every cause and obstacle that prevents young patients from adhering to insulin pumps optimally is an opportunity for intervention to improve glycemic control and, as a result, their quality of life.

## 1. Introduction

The past few decades came with a rise in the prevalence of chronic diseases in children, and one of the most prevalent chronic diseases in this age group is type 1 diabetes mellitus (T1D). T1D, which is also the most prevalent endocrine condition in children and teenagers, is brought on by an autoimmune response that targets pancreatic beta cells over the course of several months or years. As a result, insulin output declines, which can result in a partial or, more frequently, an absolute deficit. The patients must receive exogenous insulin for the remainder of their lives [1,2,3].

For patients and their caregivers, keeping the blood glucose level within the normal range can prove to be a challenge. This has served as motivation to find effective strategies to control T1D and prevent its consequences ever since insulin was discovered in 1921 [4]. The rapid development of diabetes technology over the past few decades includes continuous subcutaneous insulin infusion (CSII) as one of the most notable developments. The Diabetes Control and Complications Trial highlighted the value of rigorous therapy during childhood as it avoids and delays microvascular complications, despite the fact that the first insulin pump was created in the early 1970s [5]. This led to an increase in the popularity of the device.

Notably, following extensive research in the last decades on insulin delivery systems, the International Society for Pediatric and Adolescents Diabetes (ISPAD) advocated for insulin pump therapy for diabetes management in children of all ages [6]. Also, ISPAD recommended insulin pumps to improve glycemic controls, reduce the risk of chronic complications, and reduce episodes of hypoglycemia. In addition, real-time glucose sensors could be used to achieve better glycemic control [6].

Nevertheless, ISPAD guidelines highlighted some barriers that should be addressed in adopting device-based therapy for children with diabetes [6]. Adherence to insulin pump therapy was reported in 4% of cases, mainly due to pump wearability and anxiety. This issue could be overcome by adequate training by care providers, including progressive teaching from basic to advanced skills, and family support. Other barriers were linked to device-linked complications, such as malfunction, infusion failure, alarms, system occlusion, risk of ketoacidosis, and lipohypertrophy. Efficient communication and education offered by care providers and parents could be the solution to all these limitations, as insulin pump therapy benefits outweigh potential adverse events [6].

This paper aims to emphasize the benefits of insulin pump therapy in terms of glycemic control, as well as potential associated drawbacks. Also, we identified key factors that negatively impact adherence to insulin pumps, in order to provide solutions and potential strategies to achieve a better adherence, glycemic control, and to improve outcomes in diabetic children.

## 2. Materials and Methods

Articles from PubMed, Embase, and Science Direct databases were evaluated, using the following search terms: adherence, pump insulin therapy, children, pediatric population, and type 1 diabetes, in combination with several synonyms such as compliance, treatment adherence, pump adherence, patient dropouts, and treatment refusal. Studies included in this paper met the following criteria: full-text available online, no older than 10 years, clearly stated descriptions of samples and methodology, human subjects, children, and articles available in English. We also searched the reference lists from the included published articles to identify potentially relevant articles. In addition, both observational and randomized controlled trials (if available) were considered for inclusion in present review. Our search revealed 43 potential articles. 40 records remained after duplicate removal. Studies only describing insulin therapy adherence (insulin administered by multiple daily injections, not by means of a pump)-studies only available as abstracts-10, type 2 diabetes studies, adult studies, studies not mentioning if the pump using population was suffering from type 1 or type 2 diabetes, ongoing clinical trials, studies not taking into consideration adherence, single case reports, updates and letters to the editor, and studies failing to load were excluded. Furthermore, studies including only one type of pump were excluded due to a potential lack of objectiveness, as a possibility for that study to be supported by a specific manufacturer (Figure 1).

Selected studies were not sponsored by insulin pump manufacturers.

## 3. Results

After screening the title and abstract of retrieved references, four studies investigating insulin pump efficacy in pediatric patients were identified and analyzed. Three of them [7,8,9] were populational studies that included over 15,000 patients each, and one [10] was a prospective study conducted on almost 1000 patients (Table 1). The latter aimed to compare the frequency of microvascular complications in adolescents with type 1 diabetes undergoing CSII versus multiple daily injections (MDI) treatment variant.

Acute complications (severe hypoglycemia and diabetic ketoacidosis) were tackled by Karges et al. [9], while Sherr et al. [8], looked into metabolic control attained by using the two treatment variants. Szypowska et al. [7], concluded that HbA1c remains lower in patients using CSII. The other cited studies discussed patient adherence and satisfaction, patient education, long term efficacy of pump therapy, acute and chronic complications, quality of life, pump perception, healthcare barriers, and reasons for withdrawal from insulin pump therapy.

Notably, all studies displayed concordant results, with improved glycemic control, lower HbA1c, and a reduced risk of chronic complications linked to diabetes [7,8,9,10]. Nevertheless, these studies were observational, thus limiting the extrapolation of results to all pediatric patients. Consequently, the results should be confirmed in randomized controlled trials.

**Table 1 medicina-58-01671-t001:** Studies investigating efficiency of CSII versus MDI.

Author, Year	Type of Study	Population	Objective	Results
Szypowska et al., 2016 [7]	Cross-sectional	16,570 youth with T1D (median age 14 years)	To examine the frequency of pump usage in T1D children treated in SWEET centers and to compare metabolic control between patients treated with CSII vs. MDI.	44.4% of T1D children were treated with CSII. Both HbA1c and daily insulin dose (U/kg/d) remained decreased in children treated with CSII compared with MDI (*p* < 0.0001).
Sherr et al., 2016 [8]	Cross-sectional	54,410 youth with T1D (median age 12.1 years)	To describe differences in metabolic control and pump use in young individuals with type 1 diabetes using data collected in three multicentre registries (between 2011–2012).	Intensive treatment with an insulin pump was associated with lower HbA1c (*p* < 0.0001).
Zabeen et al., 2016 [10]	Prospective study	989 patients (aged 12–20 years; diabetes duration > 5 years) treated with CSII or MDI for >12 months.	To compare microvascular complications frequency in adolescents with type 1 diabetes treated with multiple daily injections (MDI) versus continuous subcutaneous insulin infusion (CSII)	In adolescents, CSII use is associated with lower rates of retinopathy and peripheral nerve abnormality.
Karges et al., 2017 [9]	Population-based cohort study	30,579 youth with T1D (mean age 14.1 ± 4.0)	To determine whether rates of severe hypoglycemia and diabetic ketoacidosis are lower with insulin pump therapy than insulin injection therapy in children, adolescents, and young adults with type 1 diabetes.	Compared with insulin injection therapy, insulin pump therapy was associated with lower risks of severe hypoglycemia and diabetic ketoacidosis and better glycemic control during the most recent year of therapy.

### 3.1. Adherence Obstacles

The American Diabetes Association classifies adherence barriers into three categories: patient, medication, and system barriers [11]. Good adherence to diabetes management is related to a better glycemic control. A meta-analysis that included 21 studies and 2429 adolescents concluded that better adherence is associated with better glycemic control, regardless of socioeconomic factors [12,13]. All the factors and barriers involved in achieving optimal adherence to insulin pumps are opportunities for intervention to increase glycemic control and, consequently, to increase the quality of life of young patients (Figure 2).

### 3.2. Care Provider-Patient Relationship

When discussing with young patients and their parents, the term adherence is preferred over the term compliance. Efficient communication with doctors is part of successful therapy. Furthermore, a patient actively involved in making decisions regarding health issues has an increased adherence [11,14]. Studying the SEARCH cohort of young patients with type 1 diabetes, researchers found that 48% of families felt that the providers did not discuss questions and concerns, 43% reported that communication with the diabetologist was insufficient, and 48% complained about the costs [15].

Miscommunication and misunderstanding of the recommendations negatively influence the child’s adherence to treatment. This is an excellent intervention opportunity to increase glycemic control, and practitioners should try to eliminate communication barriers. In addition, closely related to communication is diabetes education. Studies emphasize the role of continuous diabetes education delivered by a multidisciplinary team, proving that it reduces hospitalization rates, emergencies, and complications [16].

A small non-randomized study including 57 children with T1D concluded that home-based video visits are an excellent intervention to decrease HbA1c levels and increase adherence to treatment [17]. Another study documented similar results, as telehealth improved glucose control in children and adolescents with type 1 diabetes [18]. Moreover, one study reported excellent social receptivity regarding telephone and virtual visits for routine pediatric diabetes care [19]. In addition, adherence is better when patients and their parents undergo insulin pump training in a pediatric endocrinology center [20]. Therefore, telemedicine visits with both, parents and adolescents are feasible and can improve adherence to insulin pump therapy and glycaemic control. Nevertheless, these data are limited to small observational studies and large randomized controlled trials are required to confirm the results.

### 3.3. Family Role

Regarding the pediatric population and its adherence to therapy, it is crucial to consider that children are not the only managers of their treatment. Parents, other family members, teachers, and friends play an essential and decisive role in optimal diabetes management. Cross-sectional and prospective studies reported that patients with parents who are supportive, cohesive, collaborative in solving the problems, and willing to share the tasks, with an authoritative style of parenting, have good results in maintaining optimal metabolic control [6,14,21].

On the other hand, patients with disorganized families, who must face conflicts with parents who are over or under-involved in a child’s disorder, are linked with poor treatment adherence [12,22]. In addition, conflictual family relationships lead to depression among young people, while a collaborative parent-child relationship is associated with better adherence to CSII therapy and emotional stability [6]. Another aspect consists of the socio-demographic characteristics of each family. For example, lower family income, ethnic and racial issues, and numerous members lead to less parental involvement in diabetes management with negative consequences. A low income is associated with less interaction between patients and doctors [23,24].

### 3.4. Adolescence Period

Adolescents’ demand to feel autonomous rises as they mature, and, as a result, they start to take on more responsibilities related to their illness. Although the transfer in responsibility is a realistic aim, it is also associated with decreased adherence once parental supervision is reduced, particularly if parental disengagement occurs too soon. Despite the fact that children become more independent from their families once they grow up, thus impacting adherence to therapy, carbohydrate counting has a fundamental role in glycaemic control improvement [25,26,27,28,29,30].

Psychosocial, emotional, and hormonal changes that occur during adolescence have an impact on therapeutic compliance. An ideal serum glucose level must be maintained despite insulin resistance brought on by hormonal changes. According to data from medical literature, the transition to self-care throughout adolescence is when glycemic control is least likely to be achieved. Only 21% of teenagers had HbA1c levels that were within the desired range [31]. Parental participation appears to improve glycemic control and treatment adherence for diabetes, but it relies on how their children interpret this association: if the involvement is viewed as invasive, adherence declines; if it is viewed as collaborative, adherence improves [21,31].

### 3.5. Social Support

Each child or adolescent’s social interactions are a crucial component of their lives, and they have an impact on how well they use their insulin pumps. Recently, social support for young individuals with diabetes has received more attention. There are three sorts of social support discussed: informational (advice), practical assistance, and emotional support. Young diabetics’ adherence to pump therapy is likely to worsen when they must deal with their peers’ unfavorable comments about managing their diabetes. Stigma is still a real issue that harms a child’s emotional well-being. However, although quantitative research links stigma with worsening glycemic control, qualitative investigations have produced mixed outcomes [6,32,33].

## 4. Discussions

Continuous subcutaneous insulin infusion simulates physiological insulin secretion, delivering insulin in two primary ways: 24-h adjustable basal rates and prandial bolus doses. This modality of intensive treatment releases insulin in a more flexible and precise manner than multiple daily injections [34]. According to the Clinical Practice Consensus Guidelines, insulin pumps are a safe and effective method to treat type 1 diabetes in youth, regardless of age [6].

### 4.1. Benefits of Continuous Insulin Pumps

The capacity to achieve good glycemic control with decreased glucose variability, which is linked to a low risk of complications, is one of the key benefits of CSII. An HbA1c value that is close to normal is the long-term goal. While some randomized controlled studies (RCTs) demonstrated improved glycemic control, others yielded inconsistent findings and failed to have any favorable benefits on HbA1c. The patients and their parents nevertheless reported treatment satisfaction, an enhanced quality of life, and continued use of the pump despite the fact that glycated hemoglobin was not positively impacted [6].

Data collected from the SWEET registry, including 16,570 youth with type 1 diabetes mellitus, reported that the participants who used insulin pumps achieved better glycemic control with lower insulin doses than those using multiple daily injections [7]. In addition, intensive treatment with an insulin pump was associated with a lower HbA1c in a cross-sectional study that compared three transatlantic registries, including almost 55,000 participants (*p* < 0.0001) [8]. A prospective study compared HbA1c levels and microvascular complications in a cohort of 989 young people aged 12–20 treated with CSII or multiple daily injections. The HbA1c levels were similar (8.6% vs. 8.7%); however, in terms of the microvascular complications, retinopathy and microalbuminuria were at lower rates in the CSII user group [6,10].

An investigation on the effects of insulin pump therapy on children’s long-term glycemic control, frequency of severe hypoglycemia episodes, and diabetic ketoacidosis found that this treatment enhanced glycemic control; HbA1c was significantly reduced throughout the seven years of follow-up, hypoglycemia episodes were less frequent (14.7 to 7.2 events per 100 patient-years, *p* < 0.001), and the hospitalization rate for diabetic ketoacidosis was lower in pump users (2.3 vs. 4.7 per 100 patient-years, *p* = 0.003) [35]. The data analysis from the DPV database included approximately 16,460 insulin pump users and reported lower rates of hypoglycemia and diabetic ketoacidosis episodes, compared with injections users (9.55 vs. 13.97 and 3.64 vs. 4.26 per 100 patient-years) [9]. On the other hand, the actual randomized controlled trials did not succeed in proving insulin pump effects on preventing hypoglycemic episodes [4].

### 4.2. Quality of Life in Youth: Positive Aspects and Downsides

Studies found that patients’ satisfaction and quality of life are enhanced by utilizing insulin pumps based on increased freedom, flexibility in nutrition, and physical activity, even if managing type 1 diabetes is highly difficult and complex, even for adults [4]. Furthermore, a study conducted at the Department of Pediatrics of the Medical and Health Science Centre at the University of Debrecen concluded that patients’ adherence to pumps was significantly better (*p* = 0.048) compared with young patients receiving traditional treatment [36].

Regarding insulin pump therapy, numerous studies have shown that patients are satisfied, however as the medication became more widely used and accepted, difficulties also emerged. Even if it is a very effective way to deliver insulin and the benefits can exceed the disadvantages, if patients and their caregivers do not have reasonable expectations and adherence to treatment is low, the ideal glucose level cannot be reached. Although it can occasionally be overwhelming for the child and their family, a meta-analysis of 52 trials found no evidence linking CSII to a major psychological effect [4,37].

However, it is well-known that children with type 1 diabetes mellitus reported a poorer quality of life than healthy children. A recent cross-sectional study aimed to assess the psychological impact and the health-related quality of life of children using CSII or multiple daily doses of insulin. Patients using CSII had better symptom control and a better quality of life, but, on the other hand, reported more worries about pump function [38]. Wearing the device on the body all the time is another drawback. According to a review, one of the primary issues with using various devices is body image [4]. However, conflicting information is available about teenagers who use insulin pumps and their body image [39,40].

Insulin pumps use rapid-acting insulin both for basal and prandial deliveries. Taking into account the lack of long-acting or intermediate insulin depot, there is a risk of ketonemia and diabetic ketoacidosis caused predominantly by the pump failure, battery failure, dislodgement or occlusion of the infusion set, or empty insulin reservoir. Diabetic ketoacidosis can also appear when the interruption of insulin secretion is intentional, for example, when the child participates in physical competitions [11,41], with the risk of subsequent hyperglycemia and hypoglycemia that can alter neurocognitive function [12]. According to recent trials, pump dysfunction, false alarms, and infusion set malfunction are not uncommon (40–60% of users reported such accidents) [11,41].

### 4.3. Financial Issues

Regarding costs, several Eastern and Western European countries offer free insulin pumps through the health insurance system if a number of conditions are met: the patient must have health insurance, be diagnosed with T1D with supporting documentation, have a doctor’s recommendation, sign an informed consent form, and maintain a written glycemic journal to demonstrate compliance and adherence to self-care.

In Romania, two types of insulin pumps are provided: simple and sensor-augmented pumps. Sensor augmented pumps (for example, MiniMed 640G and the newer MiniMed 780G systems [42]) have special software which allows the user to customize when the pump should suspend insulin delivery in the context of hypoglycemia. The pump can automatically restart insulin delivery when the sensor detects that the glycaemic value returns to normal. Nevertheless, the situation of insulin pumps is more complex [43]. Unlike some old pumps which act only when hypoglycaemia is documented, new generation pumps can detect the descending trend of glucose levels, thus pre-emptively ceasing insulin delivery. It is of particular interest as continuous glucose sensing detects interstitial glucose levels, which presents a delay from capillary blood glucose variations. Therefore, new insulin pumps could reduce hypoglycaemia onset and severity [43].

There are four priority categories in the following order: children, adults between 18–26 years old, type 1 diabetic pregnant women, and adults older than 26 years. Despite an increase, only a small percentage of patients in the Eastern European countries use insulin pumps compared with other countries.

### 4.4. Psychological Involvement

When compared with youth without diabetes, those with diabetes are more likely to have depression, anxiety, or eating disorders. More than 15% of young individuals with diabetes reported psychological stress and anxiety, which had a negative impact on their ability to handle their blood sugar levels, according to the findings of a study whose goal was to identify predictors of care behaviors and metabolic control [6]. A disordered eating routine is more frequent among young people with diabetes than among their healthy peers (40% vs. 33%). A meta-analysis of 13 studies concluded that eating disorders are associated with poor glycemic control [44]. Diabetic adolescents with depressive symptoms (18%) reported family conflicts, and lower quality of life [45].

THR1VE! is a current randomized clinical trial that evaluates the effects of a positive psychology intervention on diabetic adolescents’ distress and glycemic control, comparing Diabetes Education and text-message-based Positive Affect intervention to a Diabetes Education control condition [46]. Neurologic and psychiatric comorbidities are associated with a higher withdrawal rate from insulin pump therapy [47]. Consequently, screening for depression, anxiety, and eating disorders and providing help should be considered a priority.

### 4.5. Sleep Quality

It is a well-known fact that sleeping routine affects cognition, school results, and good functioning throughout the day, including the management of tasks concerning diabetes control. Studies on sleep quality reported that children with type 1 diabetes get insufficient sleep, with more frequent awakenings than their healthy peers. Evidence shows that sleep disturbances and sleeping more are associated with poorer adherence to therapy [48,49].

## 5. Conclusions

Adherence to insulin pump therapy in the pediatric population is challenging and involves a multidisciplinary approach. There is much information in the literature regarding the barriers that these patients must face, but more data and involvement are needed as far as interventions are concerned. Young patients are at risk of complications associated with high morbidity and mortality rates. Prevention is critical in optimal disease management, and treatment adherence barriers must be considered to improve glycemic control. A multi-level approach could be considered to enhance adherence to insulin pump therapy. Firstly, family support and care provider-patient efficient communication should be emphasized. Secondly, according to the new studies, adherence can be improved by introducing insulin pump training in endocrinologic clinics, home-based video visits, or even engagement with positive and motivational text messages. Also, early identification of children with risk factors associated with adherence to insulin pump therapy allows for improving diabetes management.

## Figures and Tables

**Figure 1 medicina-58-01671-f001:**
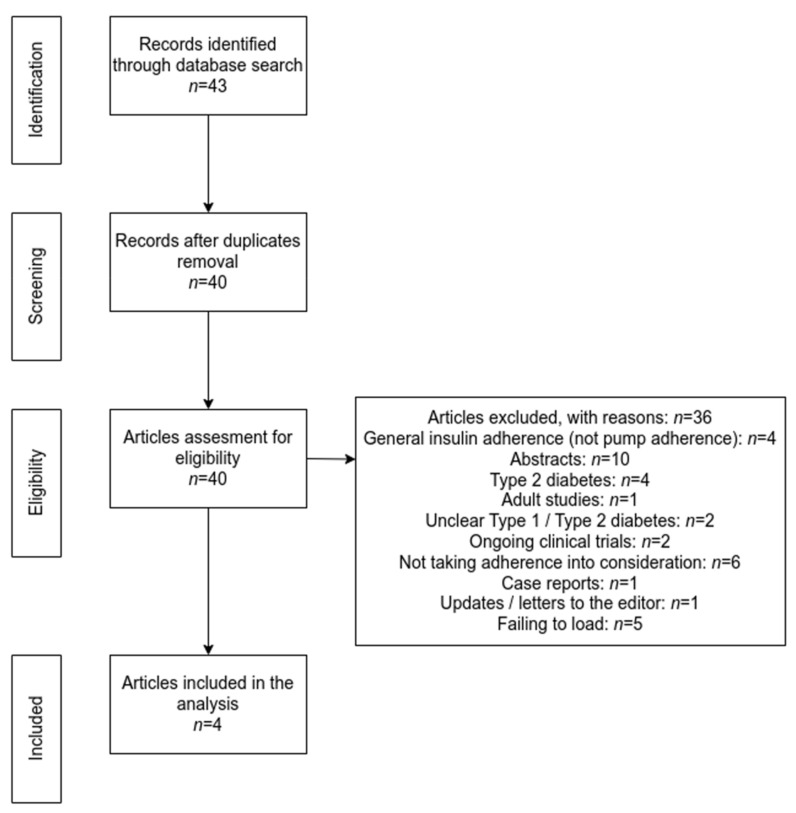
Review methodology Prisma Flowchart.

**Figure 2 medicina-58-01671-f002:**
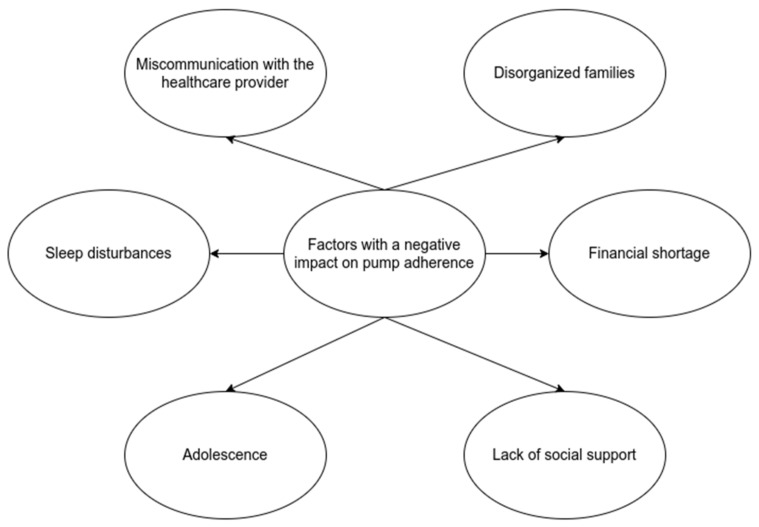
Factors with a negative impact on pump adherence.

## Data Availability

Data used in the study will be available from the corresponding authors upon request.

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
