# Peer review of "Insulin Pump Therapy Efficacy and Key Factors Influencing Adherence in Pediatric Population—A Narrative Review"

_medicina, 2022, doi:10.3390/medicina58111671_

Round 1
Reviewer 1 Report
This narrative review is interesting and updated, and the main concern is that there is no mention if the selected articles retrieved for analysis are supported by industries (insulin pumps manufacturers), condition that is relevant when considering the appropriateness and comparative advantages/disadvantages or level of satisfaction or adherence to device use.

Author Response
Thank you for the excellent point risen! The selected studies were not sponsored by insulin pump manufecturers. We included this statement in the article.

Reviewer 2 Report
The review presented is easy to read and understandable to a wide audience.
The scientific contribution is not very large but has clinical significance.
The main weakness of the review is the low number of studies on which the work was based.
It would also be convenient to include a figure or flow chart that will help to clarify the method followed in the search for articles, as well as another one describing the main results.
Author Response
Thank you for the excellent point risen!
Our search revealed 43 potential articles. 40 records remained after dupplicate removal. Studies only describing insulin therapy adherence (insulin administered by multiple daily injections, not by means of a pump)-4, studies only available as abstracts- 10, type 2 diabetes studies-4, adult studies-1, studies not mentioning if the pump using population was suffering from type 1 or type 2 diabetes-2, ongoing clinical trials-2, studies not taking into consideration adherence- 6, single case reports-1, updates and letters to the editor-1, studies failing to load-5 were excluded. Furthermore, studies including only one type of pump were excluded due to potential lack of objectiveness, as a possibility for that study to be supported by a specific manufacturer.
We included this information in the text as a Prisma flowchart and included a figure regarding the main results. We feel that our article quality wes much improved by the suggested additions. Thank you!
